# Land Transfer or Trusteeship: Can Agricultural Production Socialization Services Promote Grain Scale Management?

**Ziming Zhou, Kaihua Zhang, Haitao Wu, Chen Liu and Zhiming Yu \***

School of Business Administration, Zhongnan University of Economics and Law, Wuhan 430073, China
\* Correspondence: 202101080017@stu.zuel.edu.cn

**Abstract:** Grain Scale Management (GSM) is a crucial factor in ensuring national food security. However, in countries facing rigid resource constraints and complex land tenure relationships, the strategy of promoting large-scale grain management through land management rights transfer may not be sustainable. Therefore, based on the traditional agricultural division of labor theory, we analyze the mechanism and rationality of Agricultural Production Socialization Services (APSS) with scale characteristics to promote GSM and propose a new approach to GSM with empirical evidence from China. Using county-level panel data from Hubei province spanning from 2010 to 2021, we construct a multi-period double difference model based on the difference in the time of the establishment of pilot agricultural production socialization service counties in Hubei province. Our empirical results demonstrate the role of APSS in promoting GSM at the macro level. Specifically, the establishment of pilot counties for APSS significantly improves the level of local GSM. Furthermore, our study reveals that the degree of local financial intervention, regional industrial structure, and regional topography have heterogeneous effects on the implementation of the policy.

**Keywords:** agricultural production socialization services; grain scale management; national food security; difference-in-differences method





## 1. Introduction

Ensuring national food security is a challenge that China and all developing countries must confront directly while facing the double constraints of natural resources and business risks. In the late 1970s, China implemented the household contract responsibility system, which largely motivated farmers to grow food but also led to issues such as fragmented arable land and land management. Data from China's third agricultural census indicate that over 98% of all farm households are small farmers, and they operate 70% of the total arable land area. Although land scale management is considered to be a necessary way to improve production efficiency and reduce the risk of agricultural management [1–3], the inflexible constraints of resource endowment and the inconsistent interests of actors create constraints and bottlenecks in the process of promoting land scale management. This reality is well-reflected by the objective of land transfer in China. As of 2019, more than 555 million mu [1] of China's family-contracted land has been transferred, accounting for only about 40% of the total area of family-contracted land. Several issues such as fragmented land tenure [4], rural labor migration [5], access to the Internet [6], an incomplete land rental market [7], high contractual instability [8,9], insufficient protective technical inputs [10–12], and the security function of land [13,14] are the main reasons why the pattern of small-scale family management has not fundamentally changed, and thus the transfer of land management rights to promote large-scale grain management is unsustainable.

Classical economic theory suggests that increasing returns to scale and economic growth are driven by economies of division of labor. Unlike the path of scale management achieved through land transfer, APSS are a path of scale achieved through the division of labor in the entire agricultural production trust or semi-trust. With the continuous

upgrading of modern agricultural machinery and equipment, the divisibility of planting links and the tradability of agricultural services become more significant, leading to a more refined and specialized division of labor in grain planting. From the analysis above, it is clear that GSM cannot solely rely on land transfer and concentration to expand the scale. Instead, through APSS, small farmers can be integrated into a socialized division of labor system to achieve GSM [15] and share the economic benefits from specialized division of labor. Relevant surveys indicate [2] that APSS have the lowest average cost per mu compared to farmers' own seeds and land transfer seeds. APSS also lead to the highest yield and an average increase of more than 20% in net income. Standardized mechanized service not only brings significant scale benefits but also plays an important role in improving farmers' motivation to grow grain. Thus, APSS may be a more effective way to achieve GSM.

To promote GSM and guide small farmers to participate in APSS in an orderly manner, the government has implemented a series of policies and measures to support socialization services in agricultural production. However, in the process of policy implementation, it is important to examine whether APSS can truly promote GSM, what mechanisms are involved, and what motivates farmers to participate. This paper uses panel data from 103 county-level units in Hubei Province from 2010 to 2021 to examine the policy effects of the pilot policy of APSS in Hubei, which is considered a quasi-natural experiment in promoting GSM.

The contribution of this paper can be summarized in the following four aspects. Firstly, it analyzes the mechanism of the Agricultural Production Support System (APSS) in promoting grain security management (GSM) without land transfer based on traditional theoretical frameworks and evaluates its rationality. Secondly, through a double-difference model, it provides evidence of the policy effects of setting up pilot districts and counties for APSS in China, which can support further improvements and extensions of the APSS policy. Thirdly, it discusses the differential effects of the policy in terms of the degree of financial intervention, industrial structure, and topography, providing insights for implementing APSS policy based on local conditions. Lastly, since the APSS policy has a policy pilot nature, and there is a problem of treatment effect heterogeneity when the treatment group samples are treated at different time points, the results estimated directly using the traditional difference in difference method are biased [16]. The paper proposes a two-stage difference in difference method to enhance the robustness of the results. Overall, this paper provides valuable insights for policymakers, researchers, and stakeholders to understand and implement the APSS policy effectively.

The paper is organized as follows: Section 2 provides background information and a theoretical framework. Section 3 outlines the research methodology used in the study. In Sections 4 and 5, we present and analyze our empirical results. Finally, Section 6 concludes the paper.

## 2. Background and Assumptions

### 2.1. Policy Background

The traditional theory of agricultural division of labor starts with the assumption of land transfer and annexation as the logical starting point. The idea is that people achieve land consolidation through market-based land transfer, which leads to an expansion of the scale of agricultural cultivation [17,18]. As the scale of cultivation grows, there is a deteriorating contradiction between the scale and specialization of agricultural cultivation and the shortage and low quality of labor force [19], which leads to the use of agricultural machinery and equipment to replace manpower, thus opening up the outsourcing market of APSS [20,21]. The outsourcing market of APSS has become huge, leading to a more refined division of labor in the agricultural production chain, which in turn has made this market more mature [22,23]. This has resulted in increasing returns to scale in agriculture, which has further enhanced productivity. From the perspective of productivity, the above process reflects the evolutionary trend from inefficient human and animal power to efficient organized farm machinery power, which is a process of continuous improvement of productivity. From the perspective of production relations, the traditional theory is a

process of continuous transfer and concentration of land management rights. However, the logical starting point of the traditional theory implicitly assumes the low transaction cost of land transfer and the strong contractual spirit of small farmers. In reality, there are both external and internal constraints on land transfer in Hubei Province, which deviate from the theoretical assumptions. The external constraints are manifested in the constraints of topography and landscape and other environmental factors on land integration, and some studies point out that the heterogeneity and fragmentation of land are hard constraints on the integration of land property rights [24]. The internal constraints are manifested in the constraints of transfer of management rights by the concentration of market risk, natural risk, moral risk, and legal risk.

From the land-use data of Hubei Province [3], in the 10 years since the second land survey, the province's arable land has decreased by 8,316,800 mu, a reduction of 10.42%, and the per capita arable land is only 1.22 mu, which indicates that there is a shortage of arable land reserve resources. The contradiction of more people and less land is very serious. The aging rate of Hubei Province is 14.59%, which indicates that there is a seasonal and structural shortage of agricultural labor, and the contradiction of more people and less land is very serious. At the same time, with the escalating contradiction of planting costs, the function of grain cultivation to support the livelihood of farmers' families has been weakening, and farmers' willingness to grow grain has generally decreased [25]. Although Hubei Province has implemented many policies to support land transfer, the phenomenon of rough production and even abandonment of land has not been fundamentally solved, seriously limiting the efficiency and quality of grain production [26].

To address the challenges of land scarcity and labor shortage in Hubei Province, the government has taken steps to pilot Agricultural Production Socialization Services (APSS), which involves enhancing the capacity of APSS providers through financial subsidies. In 2017, the Hubei provincial government supported the first batch of 20 counties and cities to carry out pilot APSS, providing each county with 5–9 million yuan in support and completing a service area of 2.2 million mu. In 2018, 30 counties were supported to carry out APSS, with 7.5 million yuan in support per county, completing a total service area of over 2.37 million mu. In 2019, 22 counties were supported to carry out APSS, and in 2020, the central government allocated 570 million yuan to support 66 counties to carry out pilot APSS. In 2021, the central government allocated 250 million yuan to support 64 counties in Hubei Province to complete APSS for an area of 2.5 million mu, bringing the total number of supported counties to 77 in five years.

The government has implemented a subsidy mechanism to encourage small farmers to use mechanized services provided by APSS providers. Under this mechanism, the service provider receives a government subsidy, which is typically 30% of the market price of the service, for every acre of mechanized service provided to small farmers. In order to encourage farmers to choose mechanized services, the service provider must also reduce the price of their service. This policy is designed to incentivize market behavior rather than imposing governmental coercion on farmers. By entrusting some or all of the work aspects of production, such as cultivation, planting, prevention, and harvesting, to APSS providers, small farmers can manage their fields while the APSS providers are responsible for planting the crops. This production and management model of "managing by yourself and planting by others" not only addresses the concern of small farmers about losing their fields but also helps to solve the problem of abandoned fields. The APSS providers' responsibility to provide quality service, coupled with the government subsidy, encourages small farmers to choose mechanized services that offer higher benefits.

As of 2021, there are 30,507 agricultural production service organizations in Hubei Province, serving an area of 55,067,000 mu and 3,533,000 smallholder farmers. The area of major grain crops in the province has increased from 4,191,520 hectares in 2011 to 4,685,980 hectares in 2021, and the grain output has increased from 24,074,500 tons in 2011 to 27,643,300 tons in 2021, which has effectively promoted the development of APSS

in the province [4]. Below, we will explore the possible mechanisms and their rationality for APSS to promote GSM without land transfer.

### 2.2. Theoretical Framework and Assumptions

Firstly, it should be clarified that the entities involved in APSS include service providers, farmers, village collectives, local governments, etc. The cooperative and mutually beneficial mechanism among these entities is crucial in determining the smooth operation of APSS [27]. When each entity actively participates in APSS, it can objectively promote land consolidation. This paper will analyze the mechanisms by which each entity participates in APSS and promotes land consolidation based on their different interest demands.

For APSS providers, achieving economies of scale is the primary motivation for promoting land consolidation. The fragmented and scattered land of smallholder farmers severely limits the productivity of service providers [28] and increases their service costs. Therefore, APSS providers will use a price reduction strategy to attract small farmers with adjacent land to group together and provide mechanized services for them at a uniform time. This approach objectively promotes the consolidation of finely fragmented land into larger fields, and when the land consolidation reaches a certain scale, land scale management can be achieved. Additionally, the service provider not only averts the high risk of land rent but also prevents the excessive centralization of natural and market risks of agricultural management.

For smallholders, achieving a stable return on their land is the primary motivation for ceding their land. Small farmers are often reluctant to give up their land because it provides their livelihood. However, with the accelerated urbanization process, the returns from agricultural cultivation are much lower than those from secondary and tertiary industries, which has led to large-scale labor migration from rural to urban areas. This has resulted in the inefficient use and abandonment of a large amount of arable land, making it unprofitable for small farmers. Land transfer is constrained by various factors, making farmers cautious about conceding their land [29]. Land consolidation, as a systematic project, should fully respect the wishes of local farmers [30] and cannot be enforced by the government.

APSS is a market behavior for small farmers that does not transfer the land management right, but instead flattens the ridges of adjacent plots and blurs the boundaries of the plots. The farmers' right to income from the land is determined by the proportion of the original land to the integrated land area. This protects the farmers' income and prevents the land from being abandoned [21]. Additionally, APSS can effectively reduce natural risks such as droughts, floods, and pests, improve the production efficiency and quality of agricultural products, enhance the added value of products, and allow farmers to receive higher returns. This fully stimulates farmers to participate in socialization services [31].

As "rational" farmers, they will choose low-cost and efficient production services, as well as non-production services such as marketing, insurance, and finance to protect themselves against natural and market risks. Farmers and service providers are more likely to adjust their planting decisions and maintain consistency in planting varieties among themselves [32], thus promoting large-scale agricultural operations.

The village collective and the government serve as the bridge and link between the service providers and the farmers. The scattered characteristics of small farmers necessitate organizations to unify with service providers to enhance efficiency. Local governments need to provide targeted policy, financial, technical, and other precise assistance to APSS in order to further cultivate and grow service subjects and strengthen their service capacity. This will help to realize the organic connection between small farmers and modern agriculture, which not only helps governments fulfill their responsibilities but is also the main driving force for government participation. Village collectives can use relational network ties to promote and organize APSS. They can mobilize small farmers who work outside the village to connect with APSS providers by establishing cooperatives and other means. In this process, APSS can play a radiation-driven effect to integrate the production, processing, storage, transportation, and sales of agricultural products into a large system, maximizing

the advantages of APSS providers. This enhances the efficiency of agricultural operations and strengthens the village collective economy, which is the main driving force for village collective participation. The logical framework diagram is shown in Figure 1.

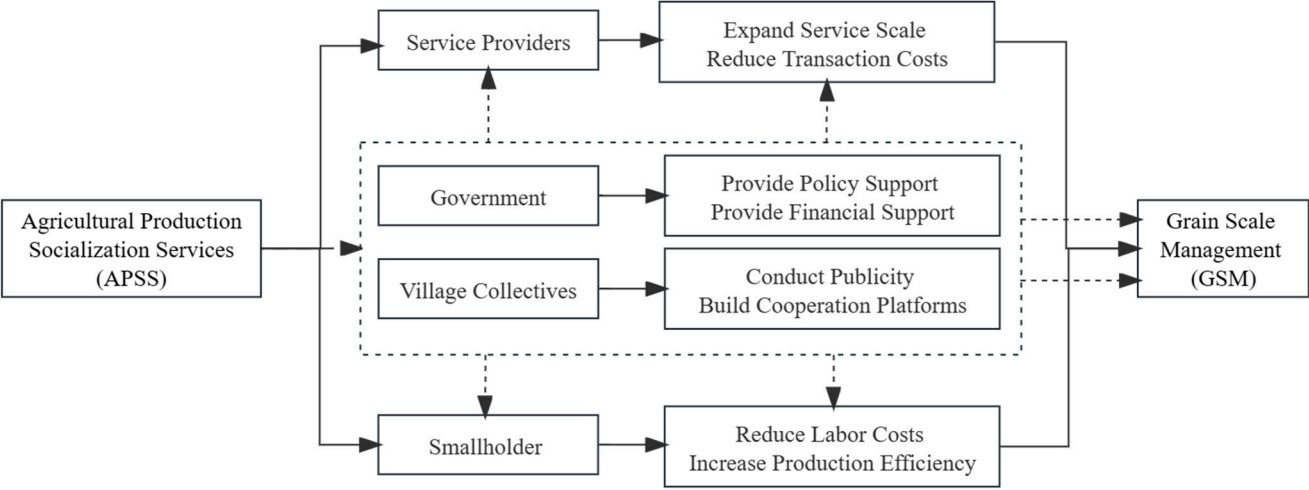

**Figure 1.** A Logical Framework for APSS to Promote GSM.

In summary, promoting GSM through APSS is a strategy that can lead to a win-win situation for all parties involved. The benefit distribution and synergistic mechanism enable APSS to function smoothly and to objectively promote land consolidation. Based on this, this paper proposes Hypothesis 1.

**Hypothesis 1.** *The pilot policy of agricultural production socialization services can effectively enhance the service level of service providers and promote grain scale management through a multi-win benefit mechanism.*

Hypothesis 1 may face a potential challenge in the implementation of the pilot policy of APSS, as the effectiveness of the policy may vary depending on the amount of financial support received from localities. The replacement of equipment of service providers and the production and expansion of agricultural enterprises require significant financial support, and financial support for agriculture plays a crucial role in improving the APSS system [33]. Although the agricultural machinery subsidy policy has played a significant role in promoting the purchase of agricultural machinery by agricultural business entities [34], the effectiveness of this incentive effect may lag due to the small and highly fragmented growth base [35]. Furthermore, when designing the promotion incentive mechanism, the government will focus on two types of indicators, namely job performance and policy burden [36]. If the policy pilot areas achieve better practical results, the pilot model will be publicized, and officials in charge of the pilot areas will increase their political performance. Based on this, officials in the pilot areas may increase financial intervention and support for the pilot policy to promote its smooth implementation, but this may lead to an overestimation of the policy effects.

Another concern is that the industrial structure reflects the direction and path of regional economic development, and different industrial structures will have an impact on local agricultural development patterns. On the one hand, the transformation of industrial structure will also create more employment opportunities for rural labor [37], and cities will absorb more foreign population and accelerate the urbanization process, which has an uncertain impact on agricultural scale operations. On the other hand, with the optimization and upgrading of industrial structure, regions will generate more financing channels and lower financing costs and thresholds, increasing the availability of financial resources for APSS providers [38]. At the same time, a constantly developing and improving industrial system will be able to

enhance the level of agricultural machinery manufacturing, provide technical support for APSS, and thus promote agricultural mechanization and large-scale operation.

Finally, from the viewpoint of objective conditions, the use of agricultural machinery and large-scale grain cultivation are seriously constrained by the topography and landscape. There is also an obvious geographical variability in the willingness of small farmers to grow food [39]. On the one hand, there are natural differences in the types of crops suitable for cultivation in different topographic regions. Differences in transportation and irrigation conditions due to geographical features can also have an important impact on farmers' willingness to grow food, and the adoption of APSS by farmers will reduce arable land abandonment [40] and promote green production behavior [41]. On the other hand, the geographic environment is a key factor affecting the service radius of APSS supply agents [42], which determines the adaptability of supply agents and APSS, and the high degree of fragmentation of arable land and inconvenient transportation in non-plain areas reduce the efficiency of agricultural machinery operation and raise the cost of agricultural machinery operation [43]. Based on this, hypothesis 2 is proposed in this paper.

**Hypothesis 2.** *Due to the differences in objective factors such as financial support, industrial structure, and geographic characteristics, there is heterogeneity in the impact of APSS on grain scale management.*

## 3. Research Method

### 3.1. Data

The empirical analysis of this paper is based on a sample of 103 county-level administrative units in Hubei province. However, due to the presence of missing values, the sample size is reduced in the empirical analysis. The time span of the sample is from 2010 to 2021, which is used to form county-level panel data in Hubei Province to assess the policy effects of the pilot APSS work in Hubei. The data used in this analysis were obtained from *the China County Statistical Yearbook*, *Hubei Rural Statistical Yearbook*, and the statistical bulletins of each county and district in past years. To ensure the comparability of sample periods, this paper takes 2010 as the base period and deflates the price-related variables. Additionally, data with too large values were processed by taking the natural logarithm for dimensionality reduction, outliers were processed by using the winsor2 command in Stata, and missing values are processed by linear interpolation or deletion.

### 3.2. Econometric Model

In this paper, we consider the pilot policy of APSS in Hubei as a quasi-natural experiment. The policy is financed by direct funding from the central government, and local governments are not required to make financial matches. To estimate the effect of APSS on GSM, we use the multi-period difference-in-difference (DID) method. This method is commonly used to assess the effect of a policy by controlling other factors constantly and testing whether there is a significant difference in the level of GSM between the treatment and control groups before and after the policy. Referring to the study of Li, P. and Lu, Y. et al. (2016) [44], we designed the model as follows.

$$Y_{it} = \beta_0 + \beta_1(treatment_i \times period_{it}) + (X_i \times f(t))'\delta + \phi treatment_i \times t + \gamma_t + \theta_i + \varepsilon_{it} \quad (1)$$

In this model, we use $i$ and $t$ to represent counties and years, respectively. $Y$ is the explanatory variable, which includes the log of annual grain sown area (LnGSA) and log of total grain production (LnTGP) in each county and district. These $treatment_i$ and $period_{it}$ are treatment group dummy variables and treatment period dummy variables, respectively, with 1 representing pilot APSS counties and 0 representing non-pilot counties. When the $i$th county has implemented the policy in year $t$, we let $period_{it} = 1$, otherwise $period_{it} = 0$. $X_i$ is the county-level characteristic, and $\gamma_t$ is the year-fixed effect. $\theta_i$ is the county fixed

effect, and $\varepsilon_{it}$ denotes the random error term. This paper uses the robust standard error of clustering at the county level.

Our concern is the coefficient of the core explanatory variable, $\beta_1$, which is the impact of the establishment of pilot APSS zones on local GSM. We also consider that the selection of pilot counties for APSS policies may not be random and is determined by various factors such as the level of regional agricultural machinery power, the level of regional socialized service market capacity, the level of regional APSS providers, the level of regional economic development, and the regional natural environment. Therefore, we controlled for the interaction term between county characteristics ($X_i$) and time trend ($t$) in the DID model, as suggested by Duflo (2001) [45]. This allows these county characteristics to have heterogeneous effects on the dependent variable across years. After controlling for the interaction term of county characteristics with time trends $(X_i \times f(t))'$ and the inclusion of a linear time trend ($treatment_i \times t$) representing the treatment group dummy variable, we relaxed the assumption of parallel trends in the DID model for the dependent variables of the treatment and control groups prior to the piloting of the APSS policy to a conditional parallel trend assumption [44]. Meanwhile, we focused on the coefficients of the core explanatory variables ($\beta_1$) to estimate the causal effect of the pilot APSS policy on GSM.

### 3.3. Variables

(1) Dependent variables. The existing literature primarily examines changes in agricultural production scale from a macroscopic point of view, focusing on the perspective of aggregate scale. At a microscopic level, it looks at the changes in farmers' resource input scale economy, with a focus on the efficiency scale [46]. Therefore, to measure the level of GSM in each county in Hubei from a macro perspective, this paper will refer to the studies of Zhang Lu, Luo Biliang (2018) [47], Jiang Song (2021) [48], Han Jiasun (2019) [46], and others. The logarithm of annual grain sown area (LnGSA) and the logarithm of total grain production (LnTGP) in each county will be selected as the dependent variables to assess the level of GSM.

(2) Core independent variables. APSS pilot area interaction term $DID_{it}$ ($DID_{it} = treatment_i \times period_{it}$). Hubei APSS pilot districts and counties were established from 2017–2021. $treatment_i$ and $period_{it}$ are treatment group dummy variables and treatment period dummy variables, respectively. When the $i$th county has implemented the policy during 2017–2021, then let $treatment_i = 1$, otherwise it is 0. When the $i$th county has implemented the policy during the $t$th year, let $period_{it} = 1$, otherwise 0.

(3) Control variables. In this paper, with reference to relevant studies, control variables are selected at five levels: first, variables such as the log of the total power of agricultural machinery (LnTPAM), log of diesel engine power (LnDEP), and log of gasoline engine power (LnGEP) are selected to control the impact brought by the power level of regional agricultural machinery. Second, we selected variables such as the log of electromechanical irrigated area (LnEIA), log of machine farming area (LnMFA), log of machine sown area (LnMSA), and log of the operating area of agricultural machinery cooperatives (LnOAAMC) variables, in order to control the impact brought by the level of regional socialized service market capacity. Third, we selected variables such as the log of the number of agricultural machinery service organizations (LnNAMSO), log of the number of agricultural machinery households year-end institutions (LnNAMH), in order to control the impact brought by the level of regional agriculture. Fourth, we select variables such as the log of Gross Regional Product (LnGDP), log of income of urban residents (LnIUR), log of income of rural residents (LnIRR), and the proportion of the added value of the secondary industry to GDP (INS) to control for the effects of regional economic development level. Fifth, we select variables such as the log of daylight hours (LnDH), log of annual precipitation (LnAP), and log annual average temperature (LnAAT) to control for the impact of the regional natural environment.

The descriptive statistics for the main variables are shown in Table 1.

**Table 1.** Summary statistics.

| Variable | Definition | N | Mean | S.D. | Min | Max |
|---|---|---|---|---|---|---|
| LnGSA | Ln Grain sown area | 1044 | 3.72 | 0.88 | 0 | 5.3 |
| LnTGP | Ln Total Grain Production | 960 | 12.39 | 0.82 | 9.94 | 14.12 |
| LnTPAM | Ln Total power of agricultural machinery | 1164 | 12.39 | 1.56 | 4.73 | 14.33 |
| LnDEP | Ln Diesel engine power | 1164 | 11.87 | 1.73 | 4.56 | 14.15 |
| LnGEP | Ln Gasoline engine power | 1152 | 8.75 | 1.59 | 2.4 | 11.22 |
| LnEIA | Ln Electromechanical irrigation area | 1119 | 9.49 | 1.87 | 3 | 12.65 |
| LnMFA | Ln Machine farming area | 1152 | 10.19 | 1.85 | 2.3 | 12.4 |
| LnMSA | Ln Machine sown area | 1140 | 8.55 | 2.47 | 0.99 | 12.02 |
| LnOAAMC | Ln Operating area of agricultural machinery cooperatives | 1035 | 8.55 | 2.55 | 0 | 12.51 |
| LnNAMSO | Ln Number of agricultural machinery service organizations | 1105 | 3.28 | 1.46 | 0 | 7.54 |
| LnNAMH | Ln Number of agricultural machinery households | 1130 | 8.75 | 1.87 | 11 | 11.73 |
| LnGDP | Ln GDP | 960 | 14.39 | 0.79 | 12.55 | 16.02 |
| LnIUR | Ln Income of urban residents | 960 | 10.03 | 0.35 | 9.28 | 10.59 |
| LnIRR | Ln Income of rural residents | 960 | 9.31 | 0.46 | 8.11 | 10.06 |
| INS | Secondary industry value added as a proportion of GDP | 948 | 0.44 | 0.14 | 0.14 | 0.89 |
| LnDH | Ln Daylight hours | 1236 | 7.39 | 0.14 | 6.9 | 7.67 |
| LnAP | Ln Annual precipitation | 1224 | 9.27 | 0.3 | 8.28 | 9.83 |
| LnAAT | Ln Average annual temperature | 1236 | 2.8 | 0.04 | 2.68 | 2.88 |

Whether each variable should be added to the model was subject to an equilibrium test. The results of the balance test in Table 2 showed that all coefficients were significant except for LnGEP, LnMSA, and LnOAAMC. This indicated that variables other than the three mentioned above can have an impact on whether a region is selected or not, and therefore need to be added to the model as selection variables. Insignificant variables were then added to the model as control variables.

**Table 2.** Balance test.

| | Control Group | Treatment Group | Unconditional Diff. |
|---|---|---|---|
| **VARIABLES** | **(1)** | **(2)** | **(3)** |
| LnTPAM | 10.13 | 0.78 | 0.368 *** |
| | [2.51] | [9.03] | (5.49) |
| LnDEP | 9.35 | 12.33 | 0.490 *** |
| | [2.62] | [0.96] | (5.96) |
| LnGEP | 7.33 | 8.99 | 0.045 |
| | [2.30] | [1.28] | (0.61) |
| LnEIA | 6.76 | 9.90 | 0.166 * |
| | [2.52] | [1.34] | (1.81) |
| LnMFA | 7.45 | 10.65 | 0.350 *** |
| | [2.72] | [1.13] | (3.79) |
| LnMSA | 5.38 | 9.05 | 0.184 |
| | [3.17] | [1.91] | (1.52) |
| LnOAAMC | 7.72 | 8.63 | 0.196 |
| | [2.31] | [2.56] | (0.79) |
| LnNAMSO | 2.03 | 3.43 | 0.195 *** |
| | [1.15] | [1.42] | (2.64) |
| LnNAMH | 6.88 | 9.03 | 0.367 *** |
| | [2.00] | [1.68] | (3.51) |

**Table 2.** *Cont.*

| VARIABLES | Control Group (1) | Treatment Group (2) | Unconditional Diff. (3) |
|---|---|---|---|
| LnGDP | 13.83 | 14.43 | 0.021 * |
| | [0.95] | [0.76] | (1.66) |
| LnIUR | 10.00 | 10.04 | 0.013 ** |
| | [0.35] | [0.35] | (2.13) |
| LnIRR | 9.08 | 9.32 | −0.011 ** |
| | [0.46] | [0.46] | (−2.05) |
| INS | 0.39 | 0.45 | −0.025 *** |
| | [0.13] | [0.14] | (−2.98) |
| LnDH | 7.37 | 7.39 | −0.023 *** |
| | [0.13] | [0.15] | (−3.00) |
| LnAP | 9.33 | 9.26 | 0.051 *** |
| | [0.30] | [0.29] | (3.60) |
| LnAAT | 2.82 | 2.80 | 0.006 *** |
| | [0.03] | [0.04] | (6.18) |

Note: This table reports summary statistics for the treatment and control groups. Columns 1 and 2 report the mean and standard deviation of the two sample groups, with the standard deviation shown in square brackets. Column 3 reports the unconditional differences between the treatment and control groups, with standard errors reported in parentheses. *** indicates 1% significance, ** indicates 5%, and * indicates 10%.

## 4. Estimation Results

### 4.1. Baseline Regression Results for the DID Model

To better analyze the impact of the pilot APSS policy, we started with a benchmark regression model, and the results are presented in Table 3 because of the multi-period DID model, it is not possible to control for treatment period dummy variables. Therefore, we use an individual and time two-way fixed effects model (TWFE) to address the endogeneity problem caused by unobservable omitted variables that do not vary over time. As discussed in the previous section, the cross-product terms of control variables, county characteristics, and treatment group dummy variables with time trends are also included to capture the heterogeneous time effects of various types of county characteristics on the dependent variable and reduce the bias caused by omitted variables. In the baseline regressions, only control variables are included in columns (1) and (4) for comparative analysis. Cross-product terms of county characteristics and time trends ($X_i \times f(t)$) are added in columns (2) and (5) to account for differences in these variables between treatment and control groups. Moreover, variables related to regional agricultural mechanization levels are included in columns (3) and (6) to control for their effects on grain sown area and grain yield.

**Table 3.** Baseline regression results.

| VARIABLES | Ln Grain Sown Area | | | Ln Total Grain Production | | |
|---|---|---|---|---|---|---|
| | (1) | (2) | (3) | (4) | (5) | (6) |
| DID | 0.131 *** | 0.094 *** | 0.052 ** | 0.070 *** | 0.049 *** | 0.055 *** |
| | (5.07) | (3.79) | (2.10) | (3.95) | (2.66) | (2.85) |
| Control variables | YES | YES | YES | YES | YES | YES |
| $X_i \times f(t)$ | | YES | YES | | YES | YES |
| $treatment_i \times t$ | | | YES | | | YES |
| County fixed effect | YES | YES | YES | YES | YES | YES |
| Year fixed effect | YES | YES | YES | YES | YES | YES |
| Observations | 1.008 | 912 | 912 | 948 | 912 | 912 |
| R-squared | 0.953 | 0.960 | 0.962 | 0.979 | 0.982 | 0.982 |

Note: The *t*-values are in parentheses. *** indicates 1% significance, ** indicates 5%. All observations are clustered at the county level.

The above passage describes the results of regression analyses that investigate the impact of a policy called APSS on grain production in districts and counties of Hubei Province. The analyses are conducted in several steps, with additional control variables added gradually in each step. The first three columns show the results of the regression analyses that investigate the impact of APSS on the area sown with grain. The coefficient for the core explanatory variable DID decreases from 0.131 to 0.052 as additional control variables are added, indicating that the effect of APSS on grain production is partially explained by other factors. However, the coefficients for all variables remain significant at least at the 0.05 level in each regression. The model in the third column, with the inclusion of all controls, has a core explanatory variable coefficient value of 0.052, indicating that, other things being equal, the area sown with grain in policy-implemented districts and counties is on average 5.2% higher compared to non-implemented districts and counties. The fourth to sixth columns show the results of the regression analyses that investigate the impact of APSS on total grain production. The coefficient for DID decreases from 0.070 to 0.055 as additional control variables are added, indicating that the effect of APSS on grain production is also partially explained by other factors. However, the coefficients for all variables remain significant at the 0.01 level in each regression. In the model in column (6), the core explanatory variable coefficient value is 0.055 after adding all the control terms, indicating that the total grain production in the policy-implemented districts and counties is on average 5.5% higher compared to the non-implemented districts and counties, other things being equal. The results of the baseline regression indicate that the pilot APSS policy in Hubei Province has a significant positive effect on GSM, with the sign of the coefficients remaining highly consistent, and the decrease in the estimated value indicates that the heterogeneous time effect of various county characteristics on the dependent variable is mitigated by the inclusion of control terms. The standard errors in the above regressions are clustered at the county level.

*4.2. Parallel Trends Test in Pre-Treatment Periods and Policy Dynamic Effects*

An important prerequisite for using the double difference method is that the treatment and control groups need to satisfy the parallel trend assumption, i.e., the temporal trends of grain sown area and total grain yield levels in the two groups remain largely consistent before the implementation of the Hubei pilot APSS. This paper draws on the methods of Jacobson et al. (1993) [49] and Li et al. (2016) [44] to identify the dynamic effects of the policy periods under the Event Study method framework. The specific model is as follows.

$$Y_{it} = \beta_0 + \sum_{k=-5}^{4} \alpha_k (treatment_i \times I_k) + (X_i \times f(t))'\delta + \phi treatment_i \times t + \gamma_t + \theta_i + \varepsilon_{it} \quad (2)$$

Among them, $I_k$ is a dummy variable, and $I_k$ is assigned to 1 if the difference between year $t$ and the year selected for the APSS pilot area is $k$, otherwise, it is taken as 0. The values of $k$ are between $-5$ and 4, and in this paper, all samples with $k \geq 4$ are classified as $k = 4$, and all samples with $k \leq -5$ are classified as $k = -5$, with $k = -1$ as the benchmark group. The rest of the meanings are the same as in Equation (1).

As can be seen from Figure 2, only the pre-policy period 4 coefficient of grain sown area is marginally significant, while the rest of the pre-policy coefficients are insignificant. In contrast, the coefficients of all interaction terms are significant from the year of policy implementation. It can be inferred that the hypothesis of parallel trend holds and there is no effective expectation before the policy implementation. Further from the dynamic effect of the policy (Table 4), both the sown area of grain and total grain production gradually increased after the base period and changed from negative coefficients to positive coefficients before the policy pilot. This paper argues that the coefficients show a significant upward trend in each year after the policy implementation, implying that there is a certain lag in the policy effect, and the existence of the time lag is related to the learning effect of local governments. When the Chinese government experiments with policies, it learns from

the practices of pilot regions and develops replicable experiences through continuous trial and error [50]. If a county achieves better results in the evaluation of policy implementation effects and is recognized by the central government and ministries, its experience will be replicated in similar regions across the country. Therefore, as the pilot period is extended and the local government's experience is enriched, the effects of the measures will only continue to emerge.

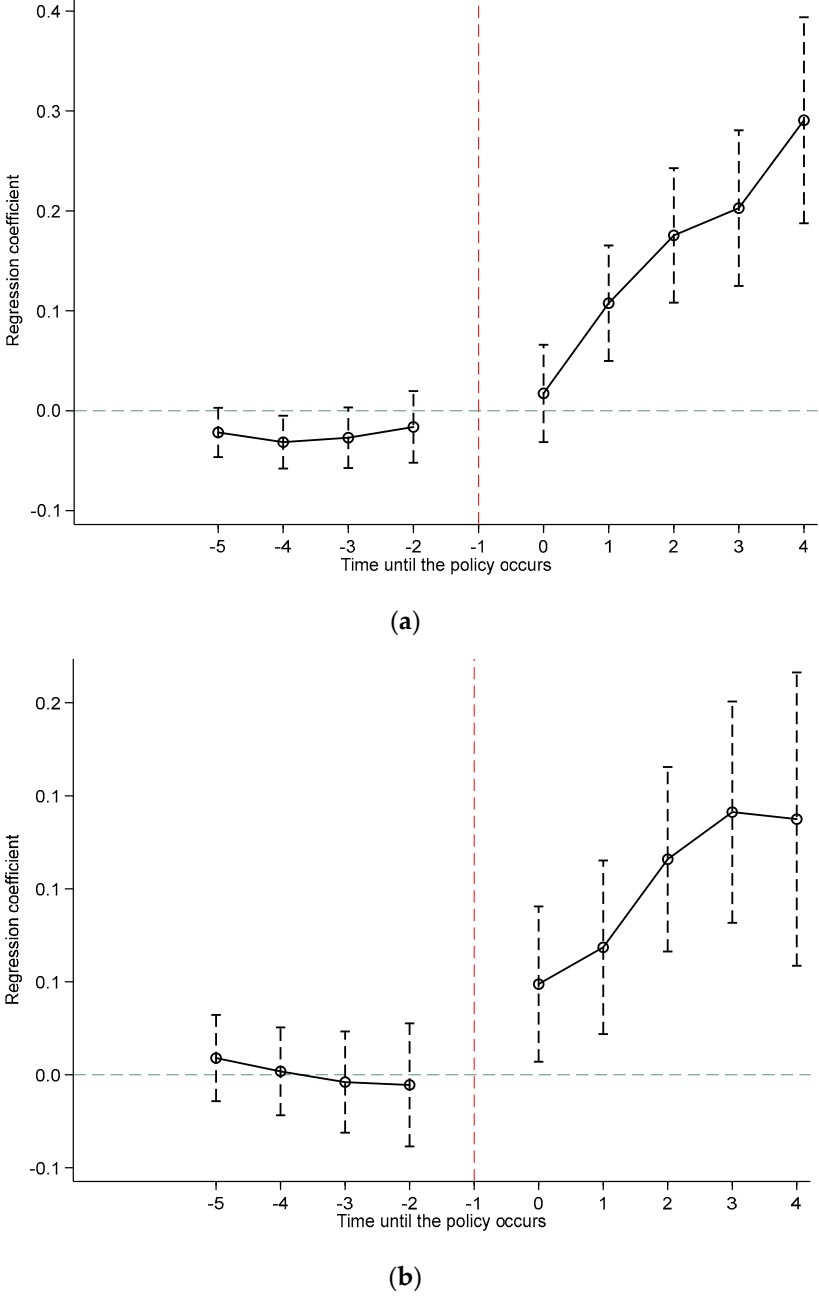

**Figure 2.** Parallel trends test in pre-treatment periods. (**a**) The grain sown area, (**b**) The Total Grain Production. Note: The graph uses the period before the policy occurred as the base period, and the coefficients for all other periods are relative to that period. The hollow points are the point estimates of the coefficients for each period, and the short vertical lines are the confidence intervals at the 95% significance level.

**Table 4.** Policy dynamic effects.

| VARIABLES | Ln Grain Sown Area (1) | Ln Total Grain Production (2) |
|---|---|---|
| treatment$*I_{-4}$ | −0.025 * | −0.001 |
| | (−1.67) | (−0.06) |
| treatment$*I_{-3}$ | −0.020 | −0.007 |
| | (−1.18) | (−0.42) |
| treatment$*I_{-2}$ | −0.010 | −0.008 |
| | (−0.47) | (−0.43) |
| treatment$*I_0$ | 0.022 | 0.047 * |
| | (0.74) | (1.89) |
| treatment$*I_1$ | 0.112 *** | 0.067 ** |
| | (3.21) | (2.39) |
| treatment$*I_2$ | 0.179 *** | 0.114 *** |
| | (4.40) | (3.83) |
| treatment$*I_3$ | 0.207 *** | 0.139 *** |
| | (4.41) | (3.90) |
| treatment$*I_4$ | 0.295 *** | 0.136 *** |
| | (4.73) | (2.85) |
| Control variables | YES | YES |
| $X_i \times f(t)$ | YES | YES |
| $treatment_i \times t$ | YES | YES |
| County fixed effect | YES | YES |
| Year fixed effect | YES | YES |
| Observations | 924 | 924 |
| R-squared | 0.963 | 0.982 |

Note: The *t*-values are in parentheses. *** indicates 1% significance, ** indicates 5%, and * indicates 10%. All observations are clustered at the county level.

*4.3. Robustness Tests*

4.3.1. Placebo Test

To further exclude the influence of other unknown factors on the selection of pilot areas and to ensure that the study findings obtained in the baseline regression are due to the pilot policy on APSS, a placebo test is required. The placebo test provides robustness to the original study findings by randomly selecting several dummy treatment groups in all samples for regressions consistent with the baseline regression. Specifically, in this paper, 1000 samples were taken from all 103 counties, 77 counties were randomly selected as the dummy treatment group in each sample, and the remaining 26 counties were regressed as the control group according to model (1). The results are shown in Figure 3. The kernel density distribution plots for both dependent variables show that the absolute value of the *t*-value of the estimated coefficients for the vast majority of the sampling is less than 1.96, the *p*-value is greater than 0.1, and the true estimates of the policy lie outside the distribution. This indicates that the policy had no significant effect on any of the 1000 random samples. Therefore, the conclusion obtained in this paper can pass the placebo test that the effect of APSS on GSM in the pilot counties is not significantly causally related to other unknown factors.

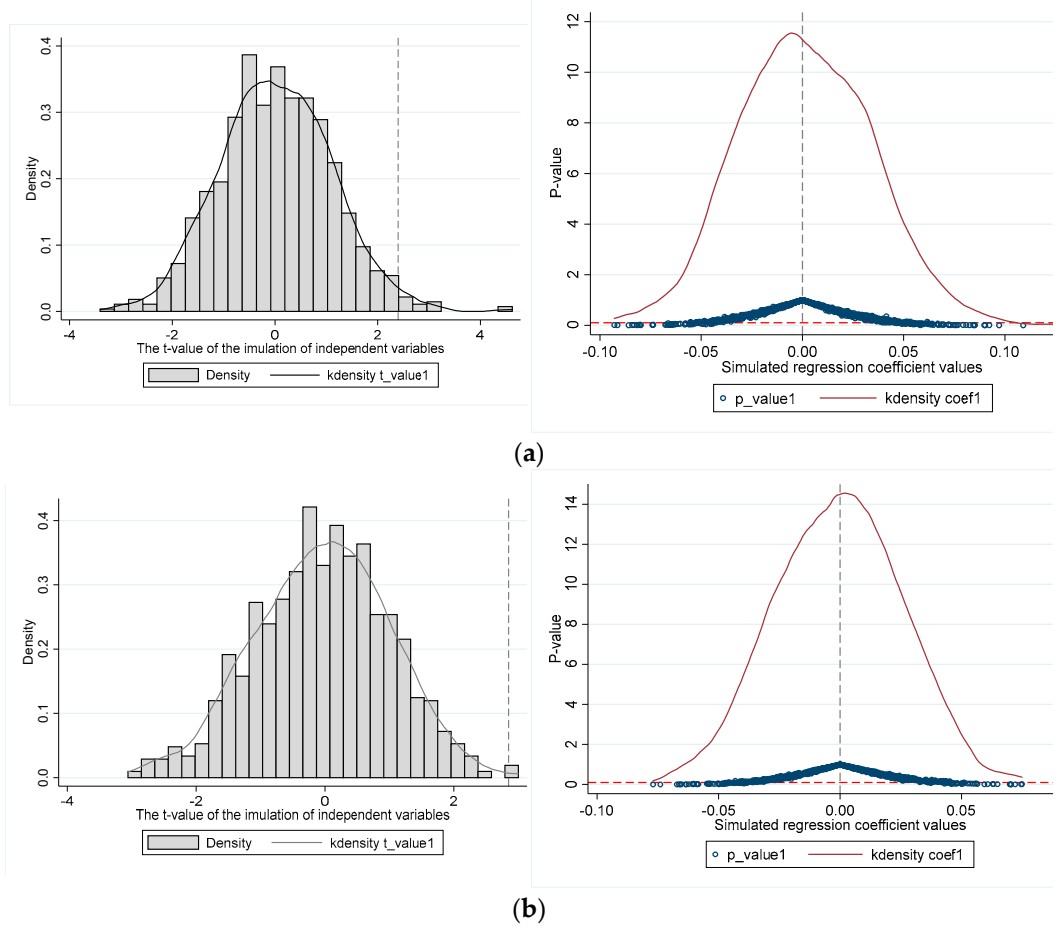

**Figure 3.** Distribution of estimated coefficients and *t*-values of falsification test. (**a**) The grain sown area, (**b**) The Total Grain Production. Note: The figure shows the cumulative distribution density of the estimated coefficients from 1000 simulations randomly assigning the pilot status to counties.

4.3.2. PSM-DID Estimation

The DID method is prone to "selectivity bias", i.e., it cannot ensure that the treatment and control groups have the same individual characteristics before the policy implementation, which is more common in the case of large sample content. The sample in this paper covers 103 county-level administrative units in Hubei province, and there may be large geographical and economic differences between the samples and a certain degree of individual heterogeneity. Therefore, this paper uses the propensity score matching method (PSM) to match the counties in the treatment and control groups using the control variables as the identifying features of the sample points. Specifically, this paper predicts the probability of each county being set up as an APSS (Logit regression) year by year using the control variables and county characteristics from the previous section. The radius matching method is then used to match the sample of pilot counties (treatment group) with the control group so that there is no significant difference between the treatment and control groups before the policy shock as much as possible to reduce the endogeneity problem caused by the self-selection bias of the counties in the pilot APSS at the time of establishment. On this basis, the net impact of the pilot APSS policy on GSM is identified using the DID approach. Since the propensity score matching method can solve the bias problem of observable covariates to the maximum extent and the double difference method can eliminate the effects of unobserved variables such as constant over time and simultaneous changes over time, the combination of the two methods can better identify the policy effects. As can be seen from columns (1) and (5) of Table 5, the coefficients of the core explanatory variables of the PSM-DID model are 0.116 and 0.066, respectively, which

do not change significantly from the baseline regression results and remain significant, indicating that APSS still significantly increase the level of regional grain sown area and total grain production, which indicates that the regression results of this paper are robust.

**Table 5.** Robust test.

| VARIABLES | Ln Grain Sown Area | | | | Ln Total Grain Production | | | |
|---|---|---|---|---|---|---|---|---|
| | **(1)** PSM-DID | **(2)** Replace the Dependent Variable | **(3)** Changing the Policy Window | **(4)** Excluding Other Policy Effects | **(5)** PSM-DID | **(6)** Replace the Dependent Variable | **(7)** Changing the Policy Window | **(8)** Excluding Other Policy Effects |
| DID | 0.116 ** | 0.077 *** | 0.051 * | 0.043 * | 0.066 *** | 0.233 *** | 0.057 *** | 0.047 ** |
| | (2.32) | (2.59) | (1.94) | (1.70) | (3.17) | (2.62) | (2.75) | (2.38) |
| Control variables | YES | YES | YES | YES | YES | YES | YES | YES |
| $X_i \times f(t)$ | YES | YES | YES | YES | YES | YES | YES | YES |
| $treatment_i \times t$ | YES | YES | YES | YES | YES | YES | YES | YES |
| County fixed effect | YES | YES | YES | YES | YES | YES | YES | YES |
| Year fixed effect | YES | YES | YES | YES | YES | YES | YES | YES |
| Observations | 147 | 912 | 615 | 860 | 912 | 615 | 860 | 912 |
| R-squared | 0.986 | 0.986 | 0.972 | 0.962 | 0.699 | 0.987 | 0.981 | 0.699 |

Note: The *t*-values are in parentheses. *** indicates 1% significance, ** indicates 5%, and * indicates 10%. All observations are clustered at the county level.

### 4.3.3. Substitution of Dependent Variables

The substitution of model-dependent variables is a commonly used robustness test in regression analysis. In this study, to test the robustness of the baseline regression results, the dependent variables of grain sown area and total grain production are replaced with rice sown surface and rice yield, respectively, while the other control variables remain consistent with the baseline regression model. The results from columns (2) and (6) of the regression analyses show that the policy effect remains significantly positive when rice sown surface and rice yield are used as the dependent variables.

### 4.3.4. Changing the Policy Window Period

The choice of the policy window period can have an impact on the estimation results of a regression analysis. A short window period can lead to insufficient sample size, which can result in biased estimation results. Conversely, a long window period may lead to a change in the sample composition of the treatment and control groups. To mitigate the effect of insufficient sample size, this study uses the full sample available in the baseline regression. In the robustness test, the starting and ending two years of the sample are excluded, and only the case of 2012–2019 is retained. The results from columns (3) and (7) of the regression analyses show that the policy effect remains significantly positive, even with a shorter window period.

### 4.3.5. Excluding Other Policy Interference

In 2009, the Wuhan Rural Property Rights Exchange was established in Hubei Province to explore and practice the separation of land ownership, contracting, and management rights. In 2012, Hubei Province took the lead in issuing the "*Regulations on Rural Land Contracting and Management in Hubei Province*", proposing to clarify ownership rights, stabilize contracting rights and liberalize management rights. In 2014, the "No. 1 central document" formally proposed that, on the basis of implementing collective ownership of rural land, farmers' contracting rights should be stabilized and land management rights should be liberalized. In November of the same year, the central government issued *the Opinions on Guiding the Orderly Transfer of Rural Land Management Rights to Develop Moderate Scale Agricultural Operations*, which formally included the theory of "separation of three rights" in the guiding ideology. Therefore, land transfer has, to a certain extent, promoted the scale operation of land and the vigorous development of new business entities, which may have an impact on regional grain sowing areas and production. The impact of land transfer on agricultural scale operation is firstly expressed in the integration of regional

farmland plots, so we counted the number of farmland plots of more than 50 mu in each county and district in Hubei Province, and calculated its proportion to the total number of regional farmland plots (per_gmjy) [5,6], and added this variable to the model to control the impact brought by the land transfer policy. The results report that the regression coefficient remains significantly positive in columns (4) and (8).

### 4.3.6. The Problem of Heterogeneous Treatment Effect of Multi-Period DID

The estimation results of the multi-period double difference method may not be robust. According to Chaisemartin and D'Haultfoeuille (2020) [16] and Goodman-Bacon (2021) [51], when the treatment group samples are treated at different start times, the conventionally estimated coefficients can be considered as a weighted average sum of the treatment effects at each time point for each treated sample. Although the sum of their weights is one, negative weights may occur. If the number of negative weights is too large, it will cause the sign of the traditional estimated coefficients to be opposite to the true estimated coefficients, resulting in unrobust regression results. In this paper, we first discuss the proportion of negative weights by drawing on Chaisemartin and D'Haultfoeuille (2020) [16], and find that β estimates the weighted sum of 232 ATTs, 183 ATTs get positive weights, 49 get negative weights, and the proportion of negative weights is only 21.12%, so this paper considers the results of the benchmark regression robust. Meanwhile, this paper refers to the two-stage double difference model proposed by Gardner (2021) [52] to obtain the regression results when considering the multi-period double difference method to deal with heterogeneity. In Table 6, the coefficients are still found to be significantly positive, so the results are robust.

**Table 6.** Regression results when considering treatment of heterogeneity.

| VARIABLES | Ln Grain Sown Area | | Ln Total Grain Production | |
|---|---|---|---|---|
| | (1) Baseline Regression | (2) Did2s | (3) Baseline Regression | (4) Did2s |
| DID | 0.052 *** | 0.087 *** | 0.055 ** | 0.048 ** |
| | (3.80) | (2.98) | (2.32) | (2.15) |
| Control variables | YES | YES | YES | YES |
| $X_i \times f(t)$ | YES | YES | YES | YES |
| $treatment_i \times t$ | YES | YES | YES | YES |
| County fixed effect | YES | YES | YES | YES |
| Year fixed effect | YES | YES | YES | YES |
| Observations | 912 | 914 | 912 | 914 |
| R-squared | 0.962 | 0.079 | 0.982 | 0.036 |

Note: The *t*-values are in parentheses. *** indicates 1% significance, ** indicates 5%. All observations are clustered at the county level.

## 5. Discussion

As previously mentioned, differences in the economic base, industrial structure, and geographic environment can lead to variations in the effectiveness of policy implementation across regions. Therefore, it is essential to conduct a heterogeneity analysis to examine the baseline regression results. In this paper, we will explore three perspectives: firstly, whether the policy effect is impacted by the level of financial intervention at the county level; secondly, whether the regional industrial structure influences the policy effect; and thirdly, whether the policy effect is influenced by topography and landscape.

### 5.1. Differences in the Degree of Financial Intervention at the County Level

In this paper, we will examine the variations in policy effects based on different degrees of county-level fiscal intervention. The proportion of local fiscal expenditure to GDP (GOV) is used to represent the degree of local government intervention in the economy [53]. The

sample is divided into three groups using 20% and 40% as the critical values, and group regressions are conducted. The results are presented in the table below.

The results are shown in Table 7. The policy effect is found to be significantly positive when the degree of local fiscal intervention is less than 20% for both sown grain area and total grain production. However, as the degree of local fiscal intervention increases, the policy effect disappears, indicating that local fiscal intervention does not promote policy implementation. This paper argues that there may be two reasons for this phenomenon. Firstly, the pilot policy of APSS in Hubei province is planned and laid out by the Hubei provincial government and allocated by the central government without the need for financial support from local finance, making the utility of county-level fiscal intervention limited. Secondly, in areas with high levels of economic development, local fiscal expenditures will be more focused on urban infrastructure construction, investment attraction, and similar activities. As a result, areas with higher degrees of fiscal intervention are more likely to be regions with developed secondary and tertiary industries, where the urban area gradually expands and the arable land area is reduced accordingly. In contrast, regions with a low degree of fiscal intervention are more likely to be agricultural regions dominated by primary industries, which well explains the positive and significant coefficients in columns (1) and (4).

**Table 7.** Differences in the degree of financial intervention at the county level.

| VARIABLES | Ln Grain Sown Area | | | Ln Total Grain Production | | |
| --- | --- | --- | --- | --- | --- | --- |
| | (1) GOV $\leq$ 20% | (2) 20% < GOV $\leq$ 40% | (3) GOV > 40% | (4) GOV $\leq$ 20% | (5) 20% < GOV $\leq$ 40% | (6) GOV > 40% |
| DID | 0.121 *** | −0.024 | −0.098 | 0.095 *** | −0.029 | 0.032 |
| | (3.50) | (−0.33) | (−1.11) | (2.80) | (−0.55) | (0.69) |
| Control variables | YES | YES | YES | YES | YES | YES |
| $X_i \times f(t)$ | YES | YES | YES | YES | YES | YES |
| $treatment_i \times t$ | YES | YES | YES | YES | YES | YES |
| County fixed effect | YES | YES | YES | YES | YES | YES |
| Year fixed effect | YES | YES | YES | YES | YES | YES |
| Observations | 537 | 291 | 86 | 537 | 291 | 86 |
| R-squared | 0.597 | 0.420 | 0.827 | 0.332 | 0.298 | 0.850 |

Note: The *t*-values are in parentheses. *** indicates 1% significance. All observations are clustered at the county level.

### 5.2. Differences in Industrial Structure

In this paper, we refer to the study of Zhang et al. (2021) [54] and use the share of regional secondary industry value added in GDP (INS) to measure the regional industrial structure. We investigate the differences in policy implementation effects under different industrial structure states. The sample is divided into four groups using 30%, 50%, and 70% as the critical values, and group regressions are conducted. The regression results are presented in Table 8.

In Table 8, for both sown area of grain and total grain production, the policy effect is not significant when the regional industrial structure is less than 50%. However, as the proportion of value added of secondary production increases, the policy effect becomes positive and significant at least at the level of 0.1 when the industrial structure is between 50% and 70%. When the proportion of value added of secondary production further increases, and the industrial structure is greater than 70%, the policy effect becomes negative, with all coefficients significant at the 0.01 level. These regression results, to a certain extent, support the previous inference that APSS requires a certain industrial base in the region to promote GSM effectively. However, a high industrial share represents a higher level of urbanization and a smaller farmland area. Thus, the policy effect can only be maximized when the industrial structure is at a moderate level, and the region can provide industrial support for APSS while meeting the area requirements for GSM.

**Table 8.** Differences in industrial structure.

| VARIABLES | Ln Grain Sown Area | | | | Ln Total Grain Production | | | |
|---|---|---|---|---|---|---|---|---|
| | **(1)** INS ≤ 30% | **(2)** 30% ≤ INS ≤ 50% | **(3)** 50% < INS ≤ 70% | **(4)** INS > 70% | **(5)** INS ≤ 30% | **(6)** 30% < INS ≤ 50% | **(7)** 50% < INS ≤ 70% | **(8)** INS > 70% |
| DID | −0.070 | −0.024 | 0.122 ** | −1.057 *** | 0.018 | 0.008 | 0.087 * | −0.135 *** |
| | (−1.44) | (−0.53) | (2.73) | (−9.30) | (0.33) | (0.21) | (1.76) | (−3.77) |
| Control variables | YES | YES | YES | YES | YES | YES | YES | YES |
| $X_i \times f(t)$ | YES | YES | YES | YES | YES | YES | YES | YES |
| $treatment_i \times t$ | YES | YES | YES | YES | YES | YES | YES | YES |
| County fixed effect | YES | YES | YES | YES | YES | YES | YES | YES |
| Year fixed effect | YES | YES | YES | YES | YES | YES | YES | YES |
| Observations | 138 | 460 | 279 | 37 | 138 | 460 | 279 | 37 |
| R−squared | 0.547 | 0.335 | 0.760 | 1.000 | 0.338 | 0.244 | 0.523 | 0.976 |

Note: The *t*-values are in parentheses. *** indicates 1% significance, ** indicates 5%, and * indicates 10%. All observations are clustered at the county level.

## 5.3. Differences in Terrain and Topography

In this paper, the proportion of the area of regional plains to the total area of regional administrative districts (pre_plains) [7] is selected to measure the differences in regional topographic landscapes and to explore the differences in the effects of policy implementation under different topographic types. In this paper, the samples are divided into four groups with 30%, 50%, and 70% as the critical values, and group regressions are conducted, and the regression results are shown in Table 9.

**Table 9.** Differences in terrain and topography.

| VARIABLES | Ln Grain Sown Area | | | | Ln Total Grain Production | | | |
|---|---|---|---|---|---|---|---|---|
| | **(1)** Pre_Plains ≤ 30% | **(2)** 30% < Pre_Plains ≤ 50% | **(3)** 50% < Pre_Plains ≤ 70% | **(4)** Pre_Plains > 70% | **(5)** Pre_Plains ≤ 30% | **(6)** 30% < Pre_Plains ≤ 50% | **(7)** 50% < Pre_Plains ≤ 70% | **(8)** Pre_Plains > 70% |
| DID | −0.034 | −0.213 ** | 0.067 | 0.096 ** | 0.040 | −0.068 | 0.079 ** | 0.064 ** |
| | (−1.01) | (−2.36) | (1.48) | (2.39) | (1.43) | (−0.74) | (2.22) | (1.99) |
| Control variables | YES | YES | YES | YES | YES | YES | YES | YES |
| $X_i \times f(t)$ | YES | YES | YES | YES | YES | YES | YES | YES |
| $treatment_i \times t$ | YES | YES | YES | YES | YES | YES | YES | YES |
| County fixed effect | YES | YES | YES | YES | YES | YES | YES | YES |
| Year fixed effect | YES | YES | YES | YES | YES | YES | YES | YES |
| Observations | 328 | 60 | 132 | 392 | 328 | 60 | 132 | 392 |
| R−squared | 0.977 | 0.992 | 0.967 | 0.969 | 0.980 | 0.990 | 0.980 | 0.984 |

Note: The *t*-values are in parentheses. ** indicates 5%. All observations are clustered at the county level.

The regression results in Table 9 indicate that for both grain sown area and total grain production, the policy effect is not significant when the percentage of local plain area is less than 30%. When the percentage of local plain area is between 30% and 50%, the policy effect is negative, with the coefficient of grain sown area significantly negative. This may be due to the problems of fragmentation and difficulty of operation in these areas, leading to the poor effect of APSS in the policy promotion process and the negative effect of farmers giving up farming and changing their livelihood strategies. When the proportion of the plain area is between 50% and 70%, the policy effect becomes positive, with the coefficient of grain sown area being not significant. When the proportion of the plain area is greater than 70%, the policy effect is significantly positive. These results indicate that APSS in Hubei are mainly effective in areas where the proportion of the plain area is relatively large. This is conducive to the scale operation of service subjects and the reduction of transaction costs of services. However, in mountainous and hilly areas, the policy effect is not satisfactory. To address this issue, we can learn from the successful experience of Shuangfeng County and other areas in Hunan Province and try to adopt hilly small farm machinery to solve practical problems and alleviate the direct impact of the geographical environment on the substitution of agricultural machinery for labor effect.

## 6. Conclusions

Promoting GSM is a pressing issue that requires urgent attention, especially in regions and countries with rigid resource constraints, complex land tenure relationships, and unsustainable land transfers, to ensure food security. This paper aims to provide these regions and countries with new paths for large-scale grain management. We constructed a DID model using the difference in the time of the establishment of pilot counties of APSS in Hubei Province. We found that APSS can promote GSM at the macro level and confirmed that this approach is a new path of scale management driven by scale services, which is different from land transfer. The study finds that (1) the establishment of pilot counties for APSS significantly increases local grain cultivation area and total grain production. After adding all controls, the sown area of grain in policy-implemented counties is, on average, 5.2% higher than that in unimplemented counties, and the total grain production in policy-implemented counties is, on average, 5.5% higher than that in unimplemented counties. (2) The findings remain robust through various identification assumption tests and robustness tests, such as ex-ante parallel trend test, placebo test, substitution of the dependent variable, two-stage double difference model, etc. (3) Further analysis shows that the policy effect is positively significant only when the degree of local fiscal intervention is less than 20% for both grain sown area and total grain production. With the increasing degree of local fiscal intervention, the policy effect disappears, indicating that local fiscal intervention does not promote policy implementation. (4) The policy effect is positive and significant only when the industrial structure is between 50% and 70% of both the sown grain area and the total grain output. This suggests that the policy effect can only be maximized when the industrial structure is at a moderate level, where localities can provide industrial support for APSS and meet the area requirements for GSM. (5) The larger the proportion of plain area, the better the policy effect, and too low a proportion of plain area will make the policy have negative effects.

For countries and regions facing resource constraints and complex land tenure relationships, agricultural-scale operations are still in the early stages of exploration. To promote scale operations, attention should be given to both the promotion of GSM through land transfer and the important role of APSS in promoting GSM. Additionally, the spatial layout of agricultural mechanization development should be optimized, and agricultural development should be promoted according to local conditions. Areas that require APSS to promote GSM need a certain industrial base, but a high proportion of industry may indicate higher levels of urbanization and less farmland area. Therefore, the industrial structure must be at a moderate level, and localities should provide industrial support for APSS while meeting the area requirements for GSM to achieve maximum policy impact. Finally, APSS are mainly concentrated in areas with a relatively large area of plains, which is conducive to the scale operation of service providers and reduces the transaction costs of services. In mountainous and hilly areas, the policy effect is not ideal, but we can learn from successful experiences, such as in Shuangfeng County in Hunan Province, and try to adopt hilly small farm machinery to solve practical problems and alleviate the effects of the geographical environment that directly affect the substitution of agricultural machinery for labor.

This paper is subject to several limitations. First, the availability of data is a challenge as the Hubei Rural Statistical Yearbook does not publish data on the scale of rural land transfer and the number of land cooperatives in the county. As a result, it is difficult to empirically test the mechanism proposed in the theoretical part of this paper. Second, the effectiveness of the study may be limited because the data collection lags behind the policy implementation, and the data used in this paper cover the period from 2010 to 2021, which may not capture the latest effects produced by the policy. Third, the measurement of the level of local agricultural scale management may also be limited. Although there are many studies on the level of agricultural scale management in academia, most of them are defined at the micro level, and there are few studies at the macro level. Therefore, this paper mainly combines the available data and references to authoritative literature to interpret the level

of local agricultural scale management. However, its scientific rigor needs to be improved. Finally, this paper mainly focuses on the services of agricultural production provided by APSS, and it does not cover the mechanisms of pre-production and post-production.

**Author Contributions:** Conceptualization, K.Z., H.W. and Z.Y.; data curation, C.L.; formal analysis, Z.Z.; funding acquisition, K.Z., H.W. and Z.Y.; methodology, Z.Y. and Z.Z.; project administration, Z.Z.; software, Z.Y. and C.L.; validation, Z.Z. and Z.Y.; writing—original draft preparation, Z.Z., Z.Y. and C.L.; writing—review and editing, K.Z. and H.W.; All authors have read and agreed to the published version of the manuscript.

**Funding:** We gratefully acknowledge the financial support from the National Social Science Foundation Major Project "Research on Multidimensional Identification and Collaborative Governance of Relative Poverty in China" (19ZDA151), the Hubei Provincial People's Government Intellectual Achievement Procurement Project "Research on Promoting the Break-through Development of Hubei County Economy" (2021HB-ZLCG-02) and the Fundamental Research Funds for the Central Universities, Zhongnan University of Economics and Law "Climate change pricing for happiness (202311004)".

**Institutional Review Board Statement:** Not applicable.

**Informed Consent Statement:** Not applicable.

**Data Availability Statement:** Data supporting reported results can be found in the open-access database of the National Bureau of Statistics. http://www.stats.gov.cn/ (accessed on 10 June 2022) the China Economic and Social Big Data Research Platform. https://data.cnki.net/ (accessed on 29 June 2022), the Department of Natural Resources of Hubei Province. https://zrzyt.hubei.gov.cn/fbjd/xxgkml/sjfb/tdzytjsj/202112/t20211217_3919353.shtml (accessed on 25 September 2022) and the Ministry of Natural Resources of China, http://www.globallandcover.com (accessed on 25 September 2022).

**Conflicts of Interest:** The authors declare no conflict of interest.

## Notes

1. Unit **mu** is a unit of land area in the Chinese municipal system. One mu is equal to sixty square meters, or about 666.667 square meters. Fifteen mu is equal to one hectare.
2. Data source: http://www.npc.gov.cn/npc/c30834/202112/e0995f9916d747e38bcc7deafda97048.shtml (accessed on 22 February 2023).
3. Data are obtained from the Department of Natural Resources of Hubei Province.https://zrzyt.hubei.gov.cn/fbjd/xxgkml/sjfb/tdzytjsj/202112/t20211217_3919353.shtml (accessed on 22 February 2023).
4. The list of pilot counties and cities for APSS in Hubei Province is obtained from the Hubei Provincial Rural Economic Management Bureau, while other relevant data are obtained from the Hubei Provincial Department of Agriculture and Rural Affairs and the Hubei Provincial Statistical Yearbook.
5. *The data bulletin of the Third National Agricultural Census of Hubei Province* states that the criteria for large-scale agricultural operations are: 50 mu or more of land planted with crops in areas where the second maturity of the year and above.
6. Land-use data of Hubei Province are obtained from Jie Yang, & Xin Huang (2022). *The 30 m annual land cover datasets and its dynamics in China from 1990 to 2021* (1.0.0) [Data set]. Zenodo. https://doi.org/10.5281/zenodo.5816591.We analyzed the number and area of farmland plots in each county and district of Hubei Province using Argis software and screened the number of farmland plots with an area greater than 50 mu.
7. Topographic and geomorphological data of Hubei Province are obtained from GlobeLand30 data (2020) of the Ministry of Natural Resources of China, http://www.globallandcover.com (accessed on 22 February 2023).

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
