# Peer review of "Land Transfer or Trusteeship: Can Agricultural Production Socialization Services Promote Grain Scale Management?"

_land, doi:10.3390/land12040797_

Round 1

Reviewer 1 Report

The submitted paper investigates if service scale operations promote increased grain production with a special focus on land transfer and trusteeship.

The paper attempts to describe ongoing policy interventions to increase grain production. The General Impression of the reviewer is that this submitted paper fits ok in this journal but needs Major Revision as at the moment it is not ready to be published in a scientifc journal.

This General Impression is underlined by the following Points.

  1. Abstract has to be rewritten and should avoid to include too specific information but at the same time lacks important e.g. methodological information. The recommendation here is to follow the basic guidelines of writing an abstract 1. A brief introduction to the topic, 2. Explanation of why the topic is important, 3. Statement about what the gap is, 4. research question/s / aim/s, 5. indication of methods and finally key message.
  2. Limitations of the paper (except for the limitation in the data) are not fully described e.g. the use of fertilisation causing arable land turned into not arable land, which of course are not Part of this paper but should be mentioned as a Limitation that is not addressed
  3. Overall the methodology is very (!) weak and needs some mayor improvements and a separate section within the paper. Throughout the paper there is inconsistency e.g. at the beginning there are three aspects mentioned but then 4 are listed. Then two hypothesis are developed which are guiding the paper etc. Hence serious improvements are required in the methodology and a separate section would create a solid base for this.
  4. Another aspects which is linked to the point above - it is not clear 1. how the logical path (line 195) was developed and 2. following how the frameworks were developed and 3. how the mechansims for socialized agricultural services to promote large-scale Train management without land Transfer established.
  5. Also a more critical comment the paper presents a very specific case (China) to derive conclusions for Asia from this case is in the reviewers opinion not appropiate - as land is very specific and how Land Administration is organised in a Country also greatly depends on the historical content and how goverments are organised and operate.

Minor comments

Overall punctuation seems to have not been checked e.g. line 25

Unit mu would need to be introduced to an international audience

Reviewer 2 Report

The manuscript brings an interesting proposal, land use and land use assessment can be an important strategic tool for decision makers and policy makers.  However, it is advisable to be precise in the handling of the concepts.

I want to start with the objective of the research, it is not clear, it is important to start the introduction section with the objective, this shows clarity of what you want in your research.

The case study analysed, Hubei province, is relevant because it can help create analytical tools that can be applied to other territories in the world, but sometimes these efforts are directed by the development models that are imposed on countries, China stands prospectively as an advanced nation in these issues, but I must comment that I am concerned about the fact of how local governments can treat Chinese small farmers, that is, behind a package of incentives there are invisible coercive measures, which may be cultural or non-cultural.  This picture I am painting should be reflected in the analysis you are doing. It is important to explain the above as it would give us a picture of the transfers in the province (e.g., Land, Various Incentives, Labour).

If the authors make a general analysis, it is important to categorise small producers, civil or governmental organisations that support agricultural practice in the region, marketing, trade in goods and services.  These structures must connect at some point forming social networks to sustain their agro-productive systems. Please explain that.

The manuscript provides a good theoretical framework but as I said before conceptually it is very poor, and that situation needs to be improved.  It is important to address the role of local knowledge and how it is articulated in the fabric of smallholder farming practice.

The authors make an effort to explain how they conducted the study, but a logical and coherent research methodology is not presented. They need to organise this section.  Contrary to the results, the authors tried to give comprehensive results which provide depth to the study.

In the conclusions section, the authors should be aware that this is the only section that should be original because they are the interpretations of the results, and it is wrong to make quotations in this section.  On the other hand, I recommend the authors to start the conclusions by responding to their research objective, if they do not know how to write conclusions, it is important that they seek help, please.

The bibliography is insufficient and obsolete; it needs to be expanded and updated, incorporating references from the year 2023.

Finally, as a final comment I would like to congratulate the authors, it is an interesting study, and I would like to see it published but it still needs more scientific work for that to happen.

Round 2

Reviewer 2 Report

I congratulate the authors, they have considered most of my observations, the manuscript has improved substantially, I would recommend this version for publication.